# Type 1 diabetes contributes to combined pulmonary fibrosis and emphysema in male alpha 1 antitrypsin deficient mice

Sangmi S. Park[1], Michelle Mai[2], Magdalena Ploszaj[2], Huchong Cai[2], Lucas McGarvey[2], Christian Mueller[3,4], Itsaso Garcia-Arcos[1,2], Patrick Geraghty[1,2]*

1 Department of Cell Biology, State University of New York Downstate Health Sciences University, Brooklyn, New York, United States of America, 2 Department of Medicine, State University of New York Downstate Health Sciences University, Brooklyn, New York, United States of America, 3 The Li Weibo Institute for Rare Diseases Research, Horae Gene Therapy Center, Worcester, Massachusetts, United States of America, 4 Department of Pediatrics, University of Massachusetts Chan Medical School, Worcester, Massachusetts, United States of America

* Patrick.Geraghty@downstate.edu

## Abstract

Type 1 diabetes (T1D) is a metabolic disease characterized by hyperglycemia and can affect multiple organs, leading to life-threatening complications. Increased prevalence of pulmonary disease is observed in T1D patients, and diabetes is a leading cause of comorbidity in several lung pathologies. A deficiency of alpha-1 antitrypsin (AAT) can lead to the development of emphysema. Decreased AAT plasma concentrations and anti-protease activity are documented in T1D patients. The objective of this study was to determine whether T1D exacerbates the progression of lung damage in AAT deficiency. First, pulmonary function testing (PFT) and histopathological changes in the lungs of C57BL/6J streptozotocin (STZ)-induced T1D mice were investigated 3 and 6 months after the onset of hyperglycemia. PFT demonstrated a restrictive pulmonary pattern in the lungs of STZ-injected mice, along with upregulation of mRNA expression of pro-fibrotic markers *Acta2*, *Ccn2*, and *Fn1*. Increased collagen deposition was observed 6 months after the onset of hyperglycemia. To study the effect of T1D on the progression of lung damage in AAT deficiency background, C57BL/6J AAT knockout (KO) mice were used. Control and STZ-challenged AAT KO mice did not show significant changes in lung function 3 months after the onset of hyperglycemia. However, histological examination of the lung demonstrated increased collagen accumulation and alveolar space enlargement in STZ-induced AAT KO mice. AAT pretreatment on TGF-β-stimulated primary lung fibroblasts reduced mRNA expression of pro-fibrotic markers *ACTA2*, *CCN2*, and *FN1*. Induction of T1D in AAT deficiency leads to a combined pulmonary fibrosis and emphysema (CPFE) phenotype in male mice.

**Data Availability Statement:** All relevant data are within the manuscript and its Supporting information files.

**Funding:** This work was funded by grants from the Alpha 1 Foundation award numbers 493373 and 614218 (P.G), from the National Heart, Lung, and Blood Institute of the National Institutes of Health under Award Numbers R56HL148774 and R01HL148774 (I. G. A.), and NIH Grants (R01-NS088689, R01-DK098252, and R24-OD018259) and the Alpha-1 Foundation (C.M.). The content is solely the responsibility of the authors and does not represent the official views of the National Institutes of Health or the Alpha-1 Foundation. The funders had no role in study design, data collection and analysis, decision to publish, or preparation of the manuscript.

**Competing interests:** The authors have declared that no competing interests exist.

## Introduction

Type 1 diabetes (T1D) is a metabolic disorder characterized by insulin deficiency and subsequent hyperglycemia. Chronic hyperglycemia resulting from T1D can have systemic effects, damaging various organs and leading to numerous complications that contribute to morbidity and mortality of the disease [1]. Different studies show impairment in the lung of T1D patients such as increased airway resistance [2], in addition to decreased lung volumes [3, 4], pulmonary elastic recoil [5, 6], and diffusing capacity of the lung for carbon monoxide (DLCO) [7]. T1D is associated with an increased risk of several respiratory diseases such as emphysema, chronic obstructive pulmonary disease (COPD), asthma, and chronic bronchitis [8]. Diabetes is also a common comorbidity in COPD and idiopathic pulmonary fibrosis (IPF) [9, 10]. TGF-β1-induced epithelial-to-mesenchymal transition (EMT) in the lung [11] and lung tissue histopathological changes with increased inflammatory cell infiltration and thickening of the alveolar septa [12] are reported in animal models of T1D.

Emphysema and loss of the alveolar respiratory surface can occur with alpha-1 antitrypsin (AAT) deficiency. AAT is an anti-protease that is predominantly produced by the liver and secreted into the bloodstream [13]. AAT is well-known for its protective role in the lungs as it neutralizes proteolytic damage of the connective tissue components of the lung by proteases, such as neutrophil elastase [14]. Inherited mutations in the gene coding AAT, *SERPINA1*, cause AAT deficiency, in which misfolding of AAT proteins triggers their accumulation in the liver and subsequently decreased AAT concentration in the blood and lungs. This can lead to the destruction of alveolar walls in the lungs and the development of emphysema. AAT is also known to play a role in heterogeneous signaling processes not necessarily linked to its anti-protease capacity, including activating phosphatases [15], and inhibiting caspase activity [16]. In the context of infectious disease, AAT has anti-HIV [17] and rhinovirus properties [18], regulates neutrophil activation and degranulation [19, 20], is involved in dendritic cell maturation and promotes regulatory T cell (Treg) differentiation [21]. In the context of inflammatory processes, AAT can increase IL-10 and IL-1Ra release [22], minimize epithelial barrier damage, lower nitric oxide production [23], and regulate IL-8-mediated neutrophil chemotaxis [24]. Finally, in a profibrotic context, AAT decreases renal fibrosis by inhibiting TGFβ-induced epithelial-mesenchymal transition (EMT) [25].

Serum concentration and activity of AAT are lower in T1D patients and are associated with hyperglycemia and the duration of diabetes [26, 27]. AAT with augmentation therapy in young subjects prevents T1D development, prolongs islet allograft survival [28], increases insulin release capacity [29], and inhibits pancreatic B-cell apoptosis [30]. *In vivo*, studies using T1D mouse models showed that administration of AAT prolonged islet graft survival and inhibited β-cell apoptosis [28, 30]. Additionally, AAT gene therapy prevented the development of T1D and inhibited insulitis [31, 32]. *In vitro*, AAT increased insulin secretion and protected β-cells from apoptosis in rat pancreatic islets [29]. Despite the suggested role of AAT in the pathogenesis of T1D, the effect of T1D on AAT deficiency is unknown.

To better understand T1D-induced pulmonary complications, the effect of hyperglycemia in the lung was studied by examining the functional and histopathological changes at two-time points following the onset of hyperglycemia using the streptozotocin (STZ)-induced mouse model of T1D. The effect of T1D on the progression of lung damage in AAT deficiency was also investigated using STZ-induced AAT knockout (KO) mice 3 months after the onset of hyperglycemia. These findings show that STZ induces fibrotic changes in the lung and that it leads to accelerated development of combined pulmonary fibrosis and emphysema (CPFE) in the absence of AAT.

## Methods

### Ethics statement

This study was performed in strict accordance with the recommendations in the Guide for the Care and Use of Laboratory Animals of the National Institutes of Health and Institutional Animal Care and Use Committee (IACUC) guidelines and the research was conducted according to the principles of the World Medical Association Declaration of Helsinki. SUNY Downstate Health Sciences University's IACUC approved the protocol (protocol number 15–10482). Animals were monitored several times daily, and any mice exhibiting signs of distress were euthanized. Distress was defined as animals showing one or several of the following characteristics; ruffled fur, arched back, respiratory distress (e.g., gasping), weight loss greater than 20%, and any notable unusual behavior. Investigators consulted with veterinarians if animals appeared distressed. All animals were anesthetized by intraperitoneal (IP) injection of a mixture of ketamine and xylazine. As animals were sedated and paralyzed during pulmonary function testing, euthanized was confirmed by cervical dislocation and no detection of a pulse, followed by vital organ collection.

### Animal models

8–13 weeks old male C57BL/6J mice were fasted for 4 hours and intraperitoneally injected with STZ (50 mg/kg) dissolved in citrate buffer for 5 consecutive days to induce T1D. Control mice were injected with citrate buffer for 5 consecutive days. 14 days after the injection, fasting blood glucose was measured and mice that had blood glucose $\geq$ 250 mg/dl were considered diabetic. Pulmonary function testing (PFT) was performed on the mice 3 and 6 months after the induction of T1D, at which point the mice were euthanized.

AAT KO mice were previously generated by knocking out *Serpina1a-e* in C57BL/6J mice, using CRISPR-Cas9 as described in Borel et al. [33]. This is a whole-body AAT knockout animal. 8–13 weeks old male AAT KO mice were injected with STZ or citrate buffer as described above. Mice will be designated as follows throughout the paper for simplification: control mice (vehicle-injected mice), STZ mice (STZ-injected mice), AAT KO (vehicle-injected *Serpina1a-e* knockout mice), and AAT KO STZ (STZ-injected *Serpina1a-e* knockout mice).

### Blood glucose and HbA1c tests

Mice were fasted for 6 hours before measuring blood glucose. Fasting blood glucose levels were measured via tail vein venipuncture using the AlphaTRAK blood glucose monitoring system. To measure glycated hemoglobin (HbA1c), mice were fasted for 4 hours and blood was collected via submandibular bleeding. HbA1c was measured using the DCA Vantage Analyzer (Siemens Diagnostics) using whole blood.

Glucose tolerance testing was performed on mice by fasting them for 4 hours before testing. Glucose solution (250 mg/ml) was injected intraperitoneally at a dose of 2.5 g/kg body weight (i.e., 250 µl injection volume for a 20 g animal). Blood glucose was measured via tail vein venipuncture using the glucometer at 0, 15, 30, 60, 90, and 120 minutes after glucose injection.

### Pulmonary function test

Mice were fasted for 4 hours before the test. Mice were anesthetized by intraperitoneal injection of ketamine/xylazine hydrochloride solution (100/10 mg/kg; Millipore Sigma). Mice were tracheostomized and connected to the ventilator via endotracheal cannula to the Flexi-Vent System (SCIREQ) for PFTs. Mice were paralyzed with pancuronium bromide

(1 mg/kg) by intramuscular injection after initiating mechanical ventilation. PFTs were performed, as previously described [34]. The Flexivent software (FlexiWare, Version 7.6, Service Pack 5) was used to calculate respiratory system resistance, pressure-volume loops, compliance, elastance, Newtonian resistance (RN), tissue dampening (G), and tissue elastance (H), as per established the protocol [34] and outlined by the manufacturer. Forced expiratory measurements were also performed to calculate the forced expired volume of 0.1 seconds ($FEV_{0.1}$), forced vital capacity (FVC), peak expiratory flow (PEF), and forced expiratory flow at 50% of FVC ($FEF_{50}$).

## Histology

After euthanasia via cervical dislocation, bronchoalveolar lavage (BAL) was collected and the lungs were perfused with 10% formalin for pressure fixation, as previously described [35]. Lung tissues were paraffin-embedded, sectioned, and stained with Masson's trichrome (Abcam) as recommended by the manufacturer. Images of lung sections were taken at 20x magnification and collagen deposition was measured in the whole upper and lower lobes of the stained lung sections (approximately 100–150 images per lobe; 200–250 images per sample) using the Orbit Image Analysis software (https://www.orbit.bio). The same images were also scored for fibrosis using the modified Ashcroft score as outlined by Hubner et al [36]. Mean linear intercept (MLI) was quantified as an index of airspace size in the upper and lower lobes of stained lung sections to assess morphological changes in lung parenchyma associated with the presence of emphysema, as described by Crowley *et al.* [37].

## Quantitative real-time PCR

Lungs were homogenized in TRI Reagent with 1.0 mm diameter zirconia beads (Biospec Products) for 30 seconds using bead beater disruption (Minibeadbeater-16, BioSpec Products). RNA was extracted as described in the manufacturer's protocol (Direct-zol RNA miniprep kit, Zymo Research). First-strand complementary DNA (cDNA) was synthesized from 1 µg of total RNA using the High-Capacity cDNA Reverse Transcription Kit (Applied Biosystems) at 25˚C for 10 min, 37˚C for 120 min, 85˚C for 5 min, followed by a cooling step at 4˚C. qRT-PCR was performed using SYBR Green Master Mix (Applied Biosystems) with the following thermocycling program: [95˚C for 5 min, 95˚C for 15 sec, 55˚C for 1 min] x 40 times, followed by 65˚C to 95˚C (0.5˚C increment) for 5 sec (C1000 Touch Thermal Cycler, Bio Rad). *Hprt1* was used as the housekeeping gene. During RNA isolate, RNA was treated with DNase using the Direct-zol RNA miniprep kit from Zymo Research. RNA (without reverse transcriptase treatment) was tested as a genomic contamination control. 1 ug of total RNA was used for the first strand cDNA template synthesis to generate 20 µl of cDNA. cDNA was dilution by a factor of 4 and 1 µl was used in a 10 µl RT-PCR reaction. Primer sequences are provided in S1 Table.

## AAT concentration in plasma

An AAT ELISA assay was performed on mouse plasma as outlined in the manufacturer's protocol (Mouse Alpha 1-Antitrypsin ELISA kit, ICL Lab). Plasma samples were diluted (1:80,000). Data were presented as AAT concentration in mg/mL plasma. All AAT and wild-type mice were screened for plasma AAT levels to confirm that they were knockouts (see S1 Fig for an example of screening data).

## Cell culture

Primary adult human lung fibroblasts were purchased from Lonza. Fibroblasts were cultured in Fibroblast Growth Medium supplemented with 0.5 mL recombinant human insulin, 0.5 mL hFGF-B, 0.5 mL GA-1000 (Gentamicin and Amphotericin B), 10 mL FBS and antibiotics (10,000 units/mL penicillin and 10,000 μg/mL streptomycin). Fibroblast cultures were maintained in collagen I (4 μg/mL)-coated wells in a humidified atmosphere with 5% $CO_2$ at 37˚C. Fibroblast cultures were tested in passages four to seven. Fibroblasts were cultured for 2 hours in serum-free media before supplementation with active plasma purified AAT (MyBioSource, Inc.) and 5 ng/ml recombinant TGFβ (R&D Systems). RNA was extracted from fibroblast for qRT-PCR. Primer sequences are provided in S1 Table.

## Statistical analyses

The majority of the data are expressed as dot plots with the means ± S.E.M. highlighted. A comparison of groups was performed by Student's t-test (two-tailed). Experiments with more than 2 groups were analyzed by 2-way ANOVA with Bonferroni posttests analysis. p values for significance were set at 0.05 All analyses were performed using GraphPad Prism Software (Version 9).

## Results

### Low-dose STZ injections induce a restrictive pulmonary pattern

STZ mice had a 2-fold increase in fasting blood glucose and an approximately 1.5-fold increase in HbA1c when compared with their controls (Fig 1A). Hyperglycemia was sustained in STZ mice throughout the study period (6 months since the onset of hyperglycemia). Consistently, the glucose clearance rate was decreased in STZ mice, with significantly higher fasting blood glucose levels at all time points compared to vehicle control mice after the glucose challenge (S2A Fig). These data confirmed that STZ injections successfully induced a sustained T1D-like phenotype, and we proceeded to analyze pulmonary parameters.

To investigate whether STZ leads to functional changes in the lungs, we conducted PFT 3 months and 6 months after the onset of hyperglycemia. Forced expiratory volume ($FEV_{0.1}$) and forced vital capacity (FVC) were both decreased by approximately 15% (p = 0.0031 and 0.0044) in STZ mice after 3 months, and no difference was observed in FEV/FVC ratios (Fig 1B), which indicates a restrictive lung function pattern. In line with this, the Pressure-Volume (PV) loop shifted downwards and to the right in STZ mice. Inspiratory capacity (IC) and compliance decreased by 14% and 25% (p = 0.0019 and 0.0009), respectively, whereas tissue damping (G) and elastance (H) increased by 1.5-fold and 1.2-fold (p = 0.0211 and 0.0063) in STZ mice, respectively (Fig 1B).

These functional changes were chronic and persisted 6 months after the onset of hyperglycemia (Fig 1C); $FEV_{0.1}$ and FVC were still decreased by 18% (p = 0.0001 and 0.0005) in STZ mice, and $FEV_{0.1}$/FVC ratio was similar between STZ and control (Fig 1C). PV loops remained shifted down and to the right, and IC and lung compliance were decreased by 18.9% and 25.7% (p < 0.0001 and p = 0.0002), respectively in STZ mice. G and H were increased by 1.5-fold and 1.3-fold (p = 0.0034 and 0.0009), respectively in STZ mice compared to the control mice. These data were consistent with those observed after only 3 months of hyperglycemia. There was no worsening of the pulmonary conditions despite STZ mice showing arrested body weight gain at 3 months of age (S2B Fig).

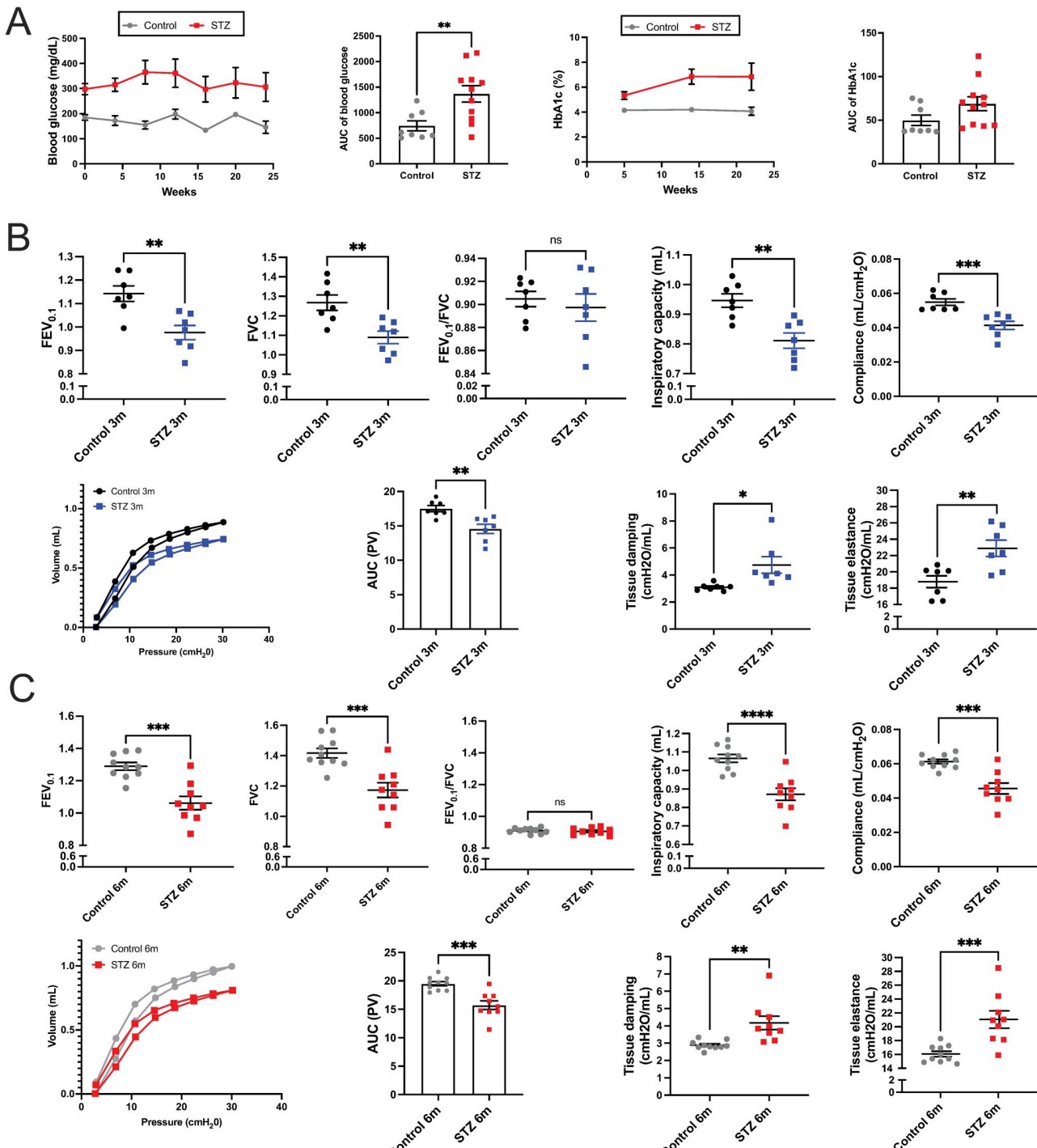

**Fig 1. Fasting blood glucose and pulmonary function tests of the vehicle and STZ-injected mice.** (A) Fasting blood glucose and glycated hemoglobin (HbA1c) were measured in mice every 4 and 8 weeks, respectively. Pulmonary function testing was performed in mice, (B) 3 months after the onset of hyperglycemia, and (C) 6 months after the onset of hyperglycemia. Data were analyzed by unpaired t-test. N = 7 to 11 animals per group. *p≤0.05; **p≤0.01; ***p≤0.001; ****p≤0.0001. AUC = Area under the curve.

## STZ mice exhibit fibrotic traits in the lung

To further investigate the functional changes in the lungs of STZ mice, we quantified collagen staining in Masson's trichrome-stained lung tissues. No collagen accumulation was observed in the wild-type mice 3 months after the onset of hyperglycemia as assessed by histological means by 2 separate methods (Fig 2A). However, mRNA expression of fibrotic markers *Ccn2* and *Fn1* was significantly increased in lung homogenates from STZ mice. By 6 months after the onset of hyperglycemia, STZ mice showed visible accumulation of collagen in the upper (3.5-fold; p = 0.001) and lower lobes (4.3-fold; p < 0.0001) of the lung, along with a significant increase in mRNA expression of *Acta2* and *Ccn2* (Fig 2B). The modified Ashcroft scoring method also confirmed elevated fibrosis in STZ mice at 6 months post STZ injections (Fig 2B). These results demonstrate the upregulation of fibrotic markers in the lungs of STZ mice as early as 3 months after the onset of hyperglycemia, which precedes collagen accumulation detectable by light microscopy by 6 months. Mean linear intercepts analysis was performed and demonstrated no emphysema development in the wild-type STZ mice at 3- or 6-months post STZ injections (S3 Fig).

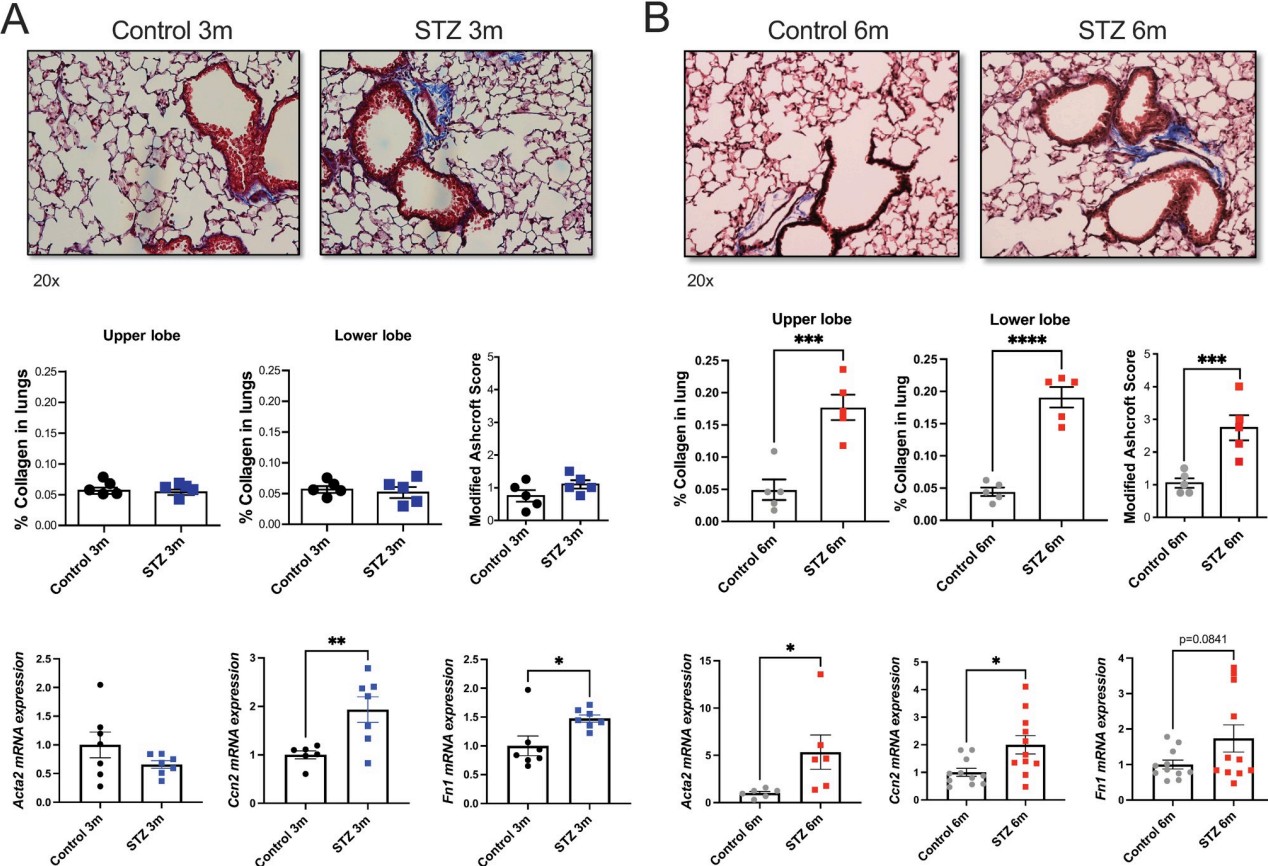

**Fig 2. Quantification of collagen and fibrotic markers in the lung of the vehicle and STZ-injected mice.** Representative images of the lung sections stained with Masson's trichrome were taken at 20x magnification. Blue staining representing collagen was quantified in the upper and lower lobes of the lung sections. Lung fibrosis was also scored using the modified Ashcroft method at each time point. *Acta2*, *Ccn2*, and *Fn1* mRNA expressions in the lung were measured by qRT-PCR. (A) 3 months after the onset of hyperglycemia and (B) 6 months after the onset of hyperglycemia. Data were analyzed by unpaired t-test. N = 5 animals per group. *p≤0.05; **p≤0.01; ***p≤0.001; ****p≤0.0001.

## Plasma AAT concentration is decreased in STZ mice and hyperglycemia is exacerbated in AAT-deficient STZ mice

T1D is associated with decreased expression of AAT, and the significance of this decrease is not understood. AAT deficiency is a risk factor for the onset of emphysema [26, 27]. Consistent with the observations in humans, plasma protein levels of AAT in STZ mice were decreased by 1.5-fold compared to the control mice (p = 0.0079) (Fig 3A) 6 months after the onset of hyperglycemia. To model the effect of T1D on the lung in AAT deficiency, we injected STZ into *Serpina1a-e* knockout (AAT KO) mice. Fasting blood glucose and HbA1c measured at several time points after the onset of hyperglycemia demonstrated that STZ AAT KO mice exhibited higher fasting blood glucose compared to STZ mice throughout the 3 months study period (Fig 3C).

## STZ AAT KO mice exhibit no significant functional changes in the lung

3 months after the validation of hyperglycemia in STZ AAT KO mice, PFTs were performed, and the results were compared to those from STZ and vehicle control mice (Fig 4). The physiology phenotype appeared to be mild and did not reach significance. AAT KO mice did not show significant differences in FEV/FVC ratios, $FEV_{0.1}$, and FVC compared to the control

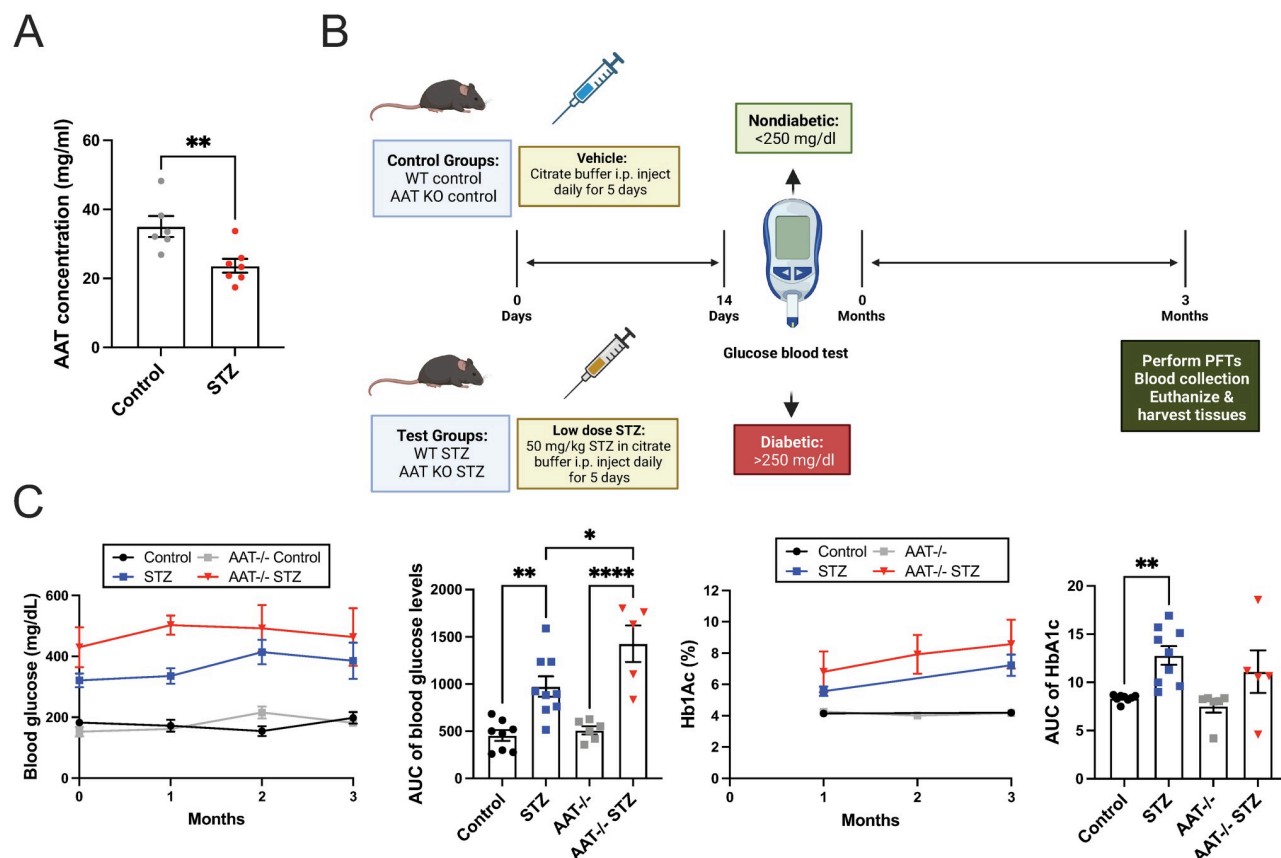

**Fig 3. Plasma AAT concentration and fasting blood glucose of vehicle and STZ-injected WT or AAT KO mice.** (A) Plasma concentration of AAT was measured in control and STZ mice. (B) Schematic showing the timeline of the experiment using vehicle and STZ-injected WT and AAT KO mice. This image was created with BioRender.com. (C) Fasting blood glucose and glycated hemoglobin (HbA1c) were measured in mice every 4 weeks. Data were analyzed by unpaired t-test and two-way ANOVA using Tukey's post hoc tests. N = 5 to 9 animals per group. *p≤0.05; **p≤0.01; ****p≤0.0001.

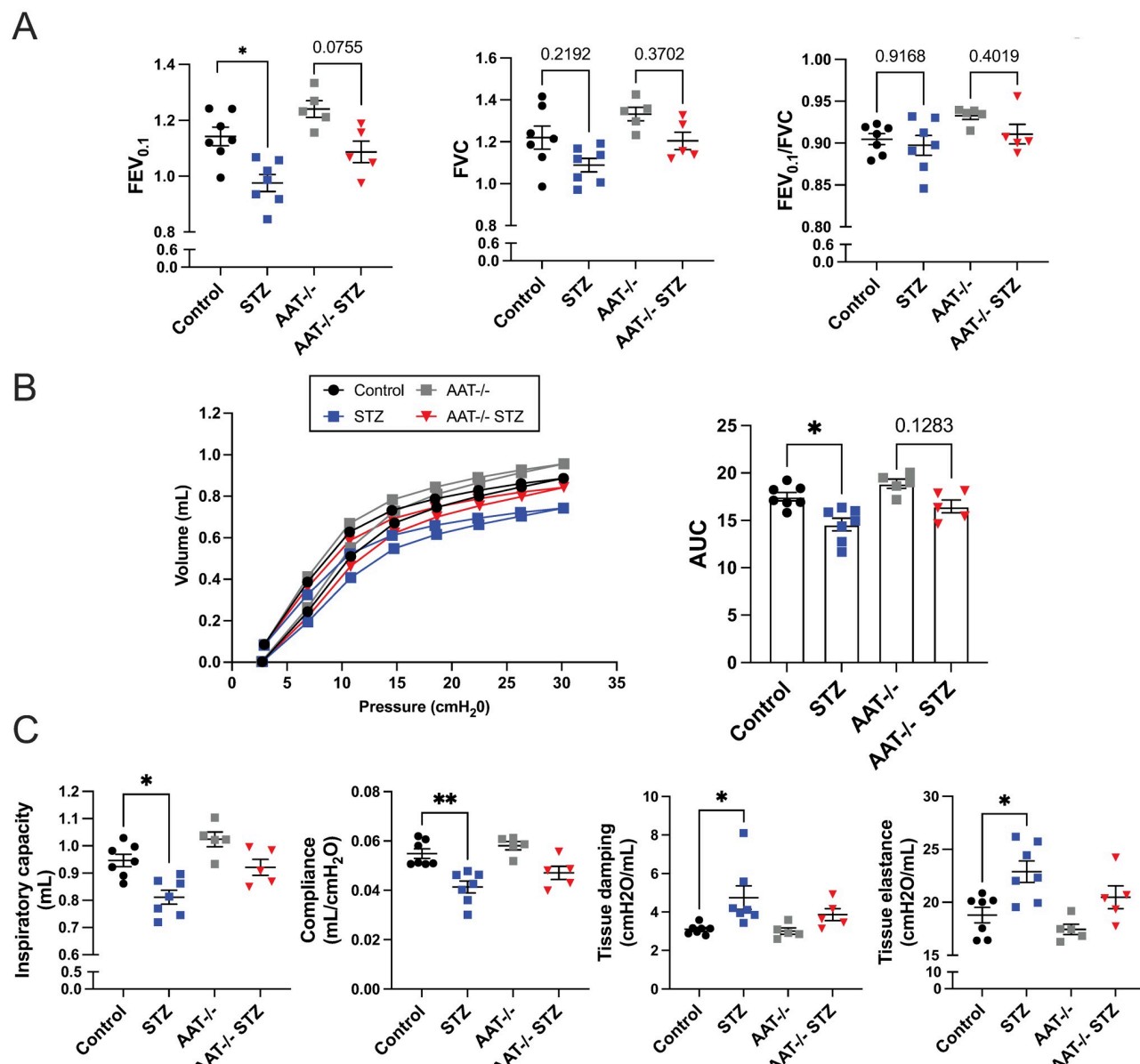

**Fig 4. PFT in the vehicle and STZ-injected WT or AAT KO mice.** PFT was performed in mice 3 months after the onset of hyperglycemia. (A) FEV0.1 and FVC measurements (B) Pressure-volume curve (AUC = Area under the curve) (C) Inspiratory capacity, compliance, tissue damping, and tissue elastance. Data were analyzed by two-way ANOVA using Tukey's post hoc tests. N = 5 to 7 animals per group. *p≤0.05; **p≤0.01.

mice (Fig 4A), and their PV loop shifted only slightly upwards and towards the left compared to the control mice (Fig 4B). Consistently, IC, compliance, G, and H were unaffected (Fig 4C).

AAT KO mice are expected to develop emphysema starting at around 35 weeks of age [33]. Since PFT was performed in the mice at 20–25 weeks of age, it can be postulated that AAT KO mice have not yet developed emphysema. Interestingly, while all of the PFT parameters in STZ AAT KO mice changed in the same direction as STZ mice, they did not reach significance (Fig 4A and 4C). PV loops of STZ AAT KO mice shifted slightly downwards and to the right compared to the AAT KO mice but this shift was not significant (Fig 4B). Overall, no obvious

functional changes were observed in AAT KO compared to the control mice and STZ did not seem to affect pulmonary function in AAT KO mice based on PFT results alone.

## STZ-induced AAT KO mice exhibit fibrotic and emphysematous histological changes in the lung

Since STZ injections were associated with a fibrotic phenotype in WT mice, we examined Masson's trichrome-stained lung sections from the AAT-deficient mouse models. STZ AAT KO mice exhibited higher collagen accumulation than AAT KO mice (~ 2-fold) in both the upper and lower lobes of the lung (p < 0.0001 and p = 0.0033). Fibrosis scoring with the modified Ashcroft method further confirmed elevated fibrosis in the STZ AAT KO mice (Fig 5B). When compared with STZ mice, STZ AAT KO mice showed upregulated *Acta2* mRNA expression in the lung (p = 0.0019) (Fig 5A and 5B). These observations suggest that the STZ injections caused a more severe increase in collagen deposition and *Acta2* gene expression in the AAT deficiency model.

Since AAT KO mice are expected to develop emphysema, MLI were analyzed in the lung sections. These 20–25 weeks-old AAT KO mice had not yet developed emphysema in either the upper or lower lobe of the lung (Fig 5C). In contrast, STZ AAT KO mice exhibited emphysema in the upper lobe of the lung with a significant increase in airway space enlargement compared to AAT KO mice. Taken together, these data suggest that the combination of AAT deficiency and T1D accelerates the progression of emphysema and induces concomitant fibrosis, leading to the development of CPFE.

## Altered TGFβ signaling observed in STZ animals *in vivo* and AAT can counter TGFβ-mediated fibroblast signaling *in vitro*

To explore inflammation responses in our mouse models, qRT-PCR and Luminex assays were undertaken. Luminex assays were performed on the plasma and BALF of wild-type animals 6 months after the onset of hyperglycemia. Plasma IL6 was decreased and CXCL5 was elevated in STZ mice at 6 months, while no changes in BALF CXCL1, CCL2, IL6, CCL20, CXCL5, and RAGE were observed (S4 Fig). TGF-β gene expression was elevated in the 3-month STZ animal group but not at the 6-month time point (Fig 6A) or in the AAT KO mice in both time points (Fig 6B).

To determine if AAT counteracted TGF-β responses, primary human lung fibroblasts were exposed to recombinant TGF-β in the presence or absence of AAT. AAT countered TGF-β-induced expression of *CCN2*, *ACTA2*, and *FN1* (Fig 6C), suggesting direct anti-fibrotic effects of AAT in fibroblasts.

## Discussion

In this study, our data show that the STZ mouse model exhibits a pattern of restrictive pulmonary defect, as indicated by decreased FEV and FVC along with decreased inspiratory capacity and compliance. These functional changes were accompanied by increased expression of fibrotic markers *Acta2* and *Ccn2* as well as accumulation of collagen in the lungs, which is consistent with other studies that suggested that T1D leads to fibrotic development in the lung [11, 12, 38]. In contrast to the changes in pulmonary function and mRNA expression of fibrotic markers in STZ mice at both 3 and 6 months after the onset of hyperglycemia, collagen staining in the lung was increased in STZ mice only at the 6-month time point. This suggests that active tissue repair and ECM deposition [39, 40] start as early as 3 months after the onset of diabetes and becomes detectable by histology at the 6-month time point.

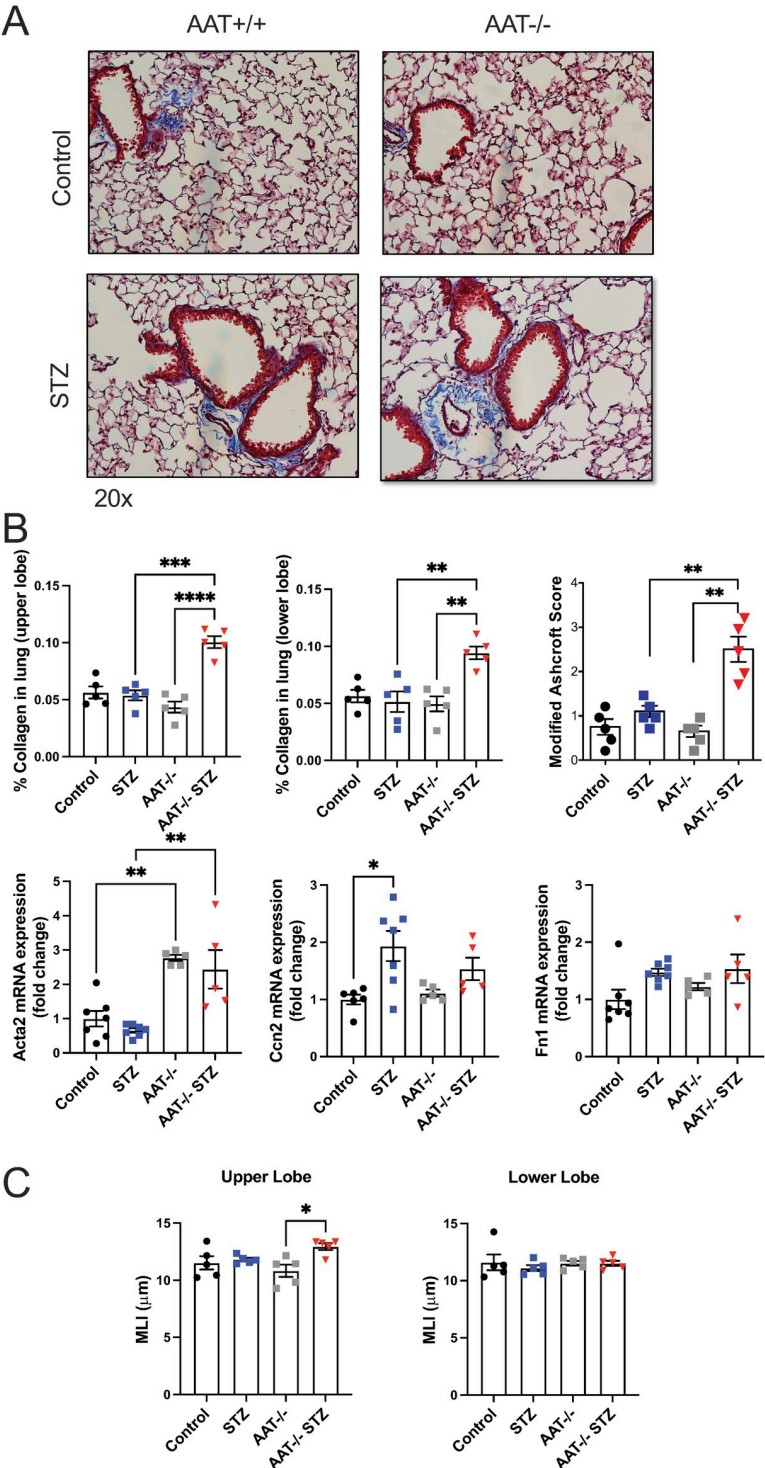

**Fig 5. Quantification of collagen and fibrotic markers in the lung of the vehicle and STZ-injected WT or AAT KO mice.** (A) Representative images of the lung sections stained with Masson's trichrome were taken at 20x magnification. (B) Blue staining representing collagen was quantified in the upper and lower lobes of the lung sections. *Acta2*, *Ccn2*, and *Fn1* mRNA expressions in the lung were measured by qRT-PCR. (C) Airspace enlargements were assessed by MLI measurements in the upper and lower lobes of the lung. Data were analyzed by two-way ANOVA using Tukey's post hoc tests. N = 5 animals per group. $*\leq0.05$; $**p\leq0.01$; $***p\leq0.001$; $****p\leq0.0001$.

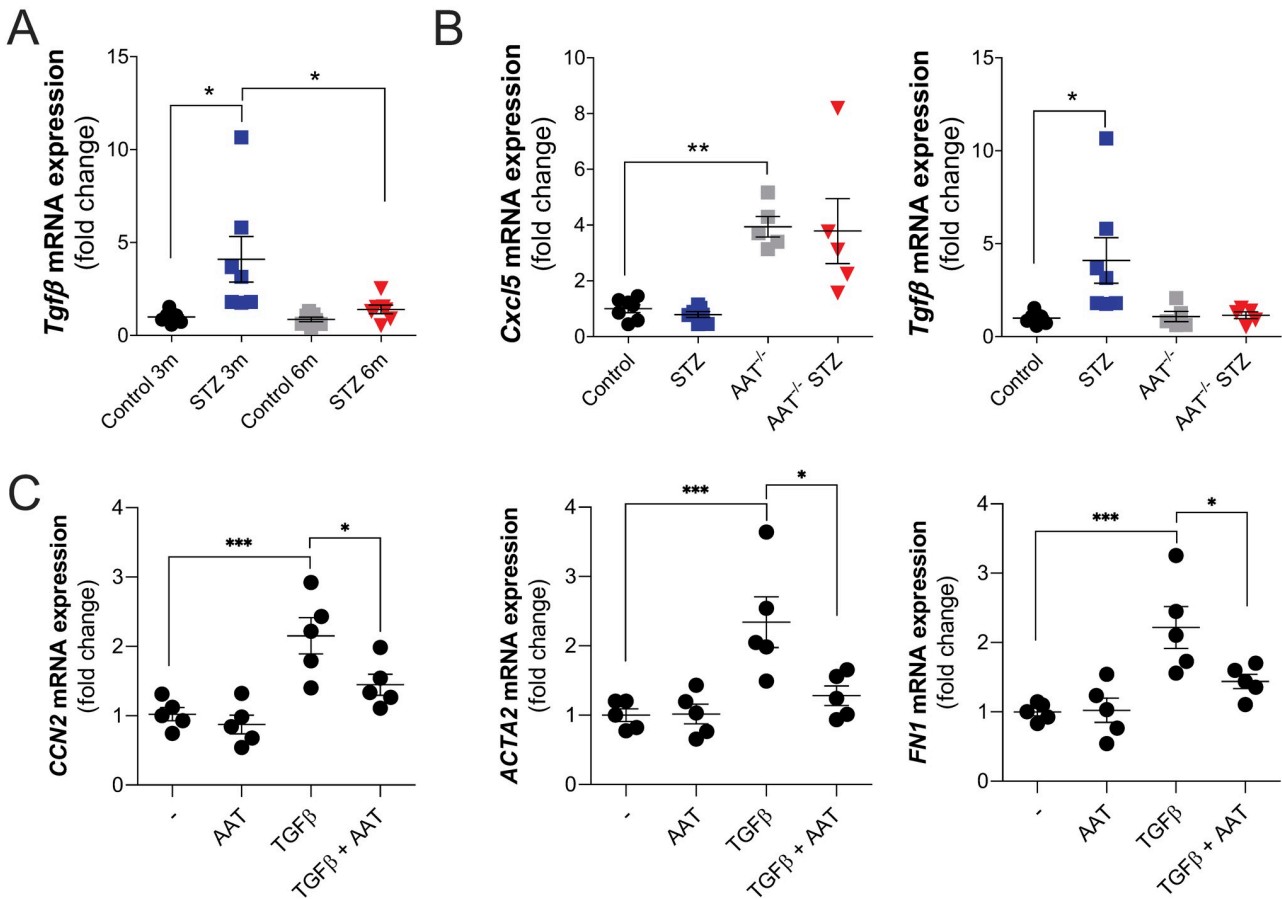

**Fig 6. Altered TGFβ signaling observed in STZ animals *in vivo* and AAT can counter TGFβ-mediated fibroblast signaling *in vitro*.** qRT-PCR was formed to determine (A) TGF gene expression changes in STZ mice at 3- and 6-months post STZ injects and for TGF and CXCL5 gene expression changes in wild-type and AAT KO mice 3-months post STZ injections. (C) qRT-PCR was performed for *CCN2*, *ACTA2*, and *FN1* changes in primary human lung fibroblasts exposed to recombinant TGF-β in the presence or absence of AAT. Data were analyzed by two-way ANOVA using Tukey's post hoc tests. *≤0.05; **p≤0.01; ***p≤0.001; ****p≤0.0001.

When combined with the AAT-deficient mouse model, STZ accelerated the development of emphysema and led to the progression of CPFE. Emphysema and pulmonary fibrosis exhibit some opposite physiologic effects. Emphysema is characterized by decreased lung elastic recoil and increased lung compliance and volume, and pulmonary fibrosis is characterized by increased lung elastic recoil and decreased lung compliance and volume. Therefore, PFT alone was insufficient to diagnose CPFE in STZ AAT KO mice [41]. However, the histology data demonstrated increased collagen accumulation in the lung of STZ-induced AAT KO mice compared to STZ mice and AAT KO control mice, suggesting that 3 months after the onset of hyperglycemia was insufficient to induce changes in the extracellular matrix (ECM) in wildtype mice but AAT deficiency accelerated this histological change, which is observed only at 6 months in STZ mice. Increased emphysema was also detected in STZ AAT KO mice compared to the control groups, which suggests that STZ accelerated the progression of emphysema. STZ AAT KO mice developed more severe hyperglycemia compared to STZ mice that express AAT, consistent with studies that show the protective role of AAT in the pathogenesis of T1D [30–32]. We also determined that elevated TGF-β expression is observed in the lungs of STZ mice and that AAT counters TGF-β signaling *in vitro* using human primary lung fibroblast cells.

Potential explanations for accelerated lung damage observed in AAT KO STZ mice include the ability of AAT to modulate inflammatory and immune responses, which is compromised in T1D models. Administration of AAT was shown to prolong islet graft survival and inhibit β-cell apoptosis [28, 30], prevent the development of T1D by preventing cell-mediated autoimmunity, and inhibit insulitis in a genetic T1D mouse model [31, 32]. AAT has anti-inflammatory properties that can suppress TNF-α and MMP-12 secretion and increase cAMP-mediated secretion of IL-10 [42–44]. TNF-α can mediate immune responses observed in the pathogenesis of T1D and inhibiting it prevented the development of T1D in nonobese diabetic (NOD) mice [45]. MMP-12 is a negative regulator of glucose metabolism and is also implicated in the development of emphysema [46, 47]. The immunosuppressive effects of IL-10 can modulate the progression of inflammatory autoimmune diseases, including T1D and COPD [48–50]. Neutrophils isolated from human T1D subjects or mice are primed to produce neutrophil extracellular traps (NETs) [51]. Neutrophils from AAT-deficient subjects have increased neutrophil responses [19, 24, 52] and AAT could counter elevated neutrophil activation, NETs formation, and degranulation. In our study, we detected elevated CXCL5 concentrations in the plasma of STZ mice and increased gene expression of *Cxcl5* in the lungs of AAT KO mice. CXCL5 is an important chemoattractant for leukocyte recruitment in mice as they lack IL8 [53]. A recent study demonstrated that monocyte-derived macrophages from ZZ AAT deficient subjects expressed higher levels of CXCL5, in addition to CXCL1 and CXCL8 [54]. AAT is also known to play a role in endothelial [55] and smooth muscle cell [56] immune responses. AAT deficiency can result in vasculitis development with elevated inflammation observed in the blood vessels [57]. In larger central arteries, the balance between elastin and collagen is important in arterial stiffness [58]. Outside of its anti-inflammatory and anti-protease functions, little is known about whether AAT influences the micro- and macrovascular of the pulmonary vasculature. Patients with AAT deficient-related COPD do have increased aortic stiffness that could lead to an increased risk of cardiovascular disease [59]. This may represent a novel area for investigation considering the impact of hyperglycemia on fibrosis and the exaggerated phenotype we observe in the AAT KO animals following STZ injections.

The role of AAT as a regulator of epithelial-mesenchymal transition (EMT) may explain the accelerated development of fibrotic phenotype in AAT KO STZ mouse lungs in our study. AAT can inhibit the Wnt canonical pathway [60], which is involved in the pathogenesis of diabetic nephropathy [61]. AAT treatment can also inhibit renal fibrosis by inhibiting TGF-β-mediated EMT [25]. Our *in vitro* work suggests that AAT could counter TGF-β-mediated signaling. However, we were not able to detect altered TGF-β gene expression in the AAT KO STZ animals. This may be due to only sampling at the 3-month time point. Additional sampling at other time points may be beneficial to further elucidate the mechanism for the fibrotic profile observed in this study.

One of the limitations of this study is that our AAT deficiency model is a knockout mouse, different from AAT deficiency in humans caused by mutations in the *SERPINA1* gene and intrahepatic accumulation of misfolded AAT. Null mutations of the *SERPINA1* gene are reported in humans but are very rare [62]. Second, several studies suggest that female mice are resistant to STZ-induced diabetes, therefore we focused on utilizing male mice [63]. This current study can only account for the results observed in male animals. Confirmation of our findings will require testing in female animals. The T1D literature suggests that female mice are more resistant to streptozotocin than male animals [64–68]. A recent study demonstrated that the resistance of female mice to STZ-induced diabetes can be overcome by increasing the dose of STZ [69]. Therefore, it may be possible to undertake a female study with this modified dosing. We also started our experiments when animals were between 8–13 weeks, which may not exactly match the age for onset of T1D in humans. Additionally, the maturational rate of

mice does not linearly correlate with humans and some suggest that maturation occurs in mice 150 times faster during the first month of life and 45 times faster over the next five months [70]. Mature adult mice range in age from 3–6 months, which is predicted to be equivalent to humans ranging from 20–30 years. Therefore, 8–13-week-old mice used in our study could be deemed to be close to 15–20 years old in human aging.

Furthermore, the impact of hyperglycemia was previously observed to reduce serum AAT concentration and activity in human T1D subjects [27], but AAT levels typically increase with inflammation and are viewed as an acute-phase reactant [71]. In T1D, there are higher levels of systemic inflammation. Therefore, the reduced AAT levels observed in our model are consistent with human AAT levels in T1D. Other cells and tissues produce AAT but the majority comes from the liver. Perhaps hyperglycemia has some long-term effects on liver function that could contribute to lower AAT levels. Equally, it is unclear how hyperglycemia impacts the activity of AAT. These are areas of interest for future studies. We cannot rule out the possibility that STZ itself could induce liver damage and reduce AAT production. The exact mechanism for the development of CPFE in the AAT KO animals remains unknown, but this will be an important area of future investigation. AAT's regulation of TGF-β signaling may be a critical event but further work is required to elucidate the exact mechanism. Other investigators have expressed the human non-mutated AAT gene, by a recombinant adeno-associated virus, in nonobese diabetic (NOD) mice and observed significantly reduced insulitis and prevented the development of hyperglycemia [31]. It would be of interest to either express a mutated version of human AAT or administer AAT to the AAT KO mice and observe glucose levels and pulmonary function following STZ-induced hyperglycemia. Finally, we only tested the AAT KO mice 3 months after STZ injections. Additional time points should be explored to determine the acute and long-term impact of AAT deficiency and hyperglycemia on disease severity and earlier inflammation or TGFβ signaling.

Altogether, our study shows that induction of T1D by STZ in mice is associated with the development of pulmonary fibrosis and that the combination of STZ injection and AAT KO accelerates the progression of pulmonary fibrosis as well as emphysema.

## Supporting information

**S1 Fig. Plasma AAT concentration for the screening of AAT KO mice.** (A) Plasma concentration of AAT was measured in control and STZ mice using a commercially available ELISA. The dotted lines denote the limit of detection of the assay. The data are expressed as dot plots with the means ± S.E.M.
(TIF)

**S2 Fig. Measurement of body weight changes and glucose tolerance testing in mice.** (A) Mice were challenged with glucose (2 g/kg, i.p.), and fasting blood glucose was measured at 0, 15, 30, 60, 90 and 120 min after the challenge. Glucose challenge was performed at 2 months and 5 months since STZ or citrate buffer injections. (B) Body weights of the mice were measured weekly and % body weight change was calculated. (A-B) The bar graphs show the area under the curve (AUC) of curves. Data were analyzed by unpaired t-test. N = 8–18. *$p \leq 0.05$, **$p \leq 0.01$.
(TIF)

**S3 Fig. Measurement of mean linear intercept for the assessment of emphysematous changes in the lungs of STZ mice.** Mean free distance in the airspace was assessed by mean linear intercept measurements in the upper and lower lobes of the lung in control or STZ mice exposed to RA or CS for 6 months. Data were analyzed by two-way ANOVA using Tukey's

post hoc tests. N = 5/group.
(TIF)

**S4 Fig. Inflammatory changes in the plasma and BAL of STZ mice.** Protein concentrations of CXCL1, CCL2, IL-6, CCL20, CXCL5, and RAGE were measured by Luminex multiplex assay in the A) plasma and B) BAL samples from control and STZ mice. Data were analyzed by Student's t-test. *p≤0.05; **p≤0.01; ***p≤0.001.
(TIF)

**S1 Table. Primer sequences used for qRT-PCR.**
(DOCX)

## Acknowledgments

The authors would like to thank the Pulmonary Division of SUNY Downstate Health Sciences University for its support.

## Author Contributions

**Conceptualization:** Sangmi S. Park, Itsaso Garcia-Arcos, Patrick Geraghty.

**Data curation:** Sangmi S. Park, Michelle Mai, Magdalena Ploszaj, Huchong Cai, Lucas McGarvey, Patrick Geraghty.

**Formal analysis:** Sangmi S. Park, Lucas McGarvey, Patrick Geraghty.

**Funding acquisition:** Christian Mueller, Itsaso Garcia-Arcos, Patrick Geraghty.

**Investigation:** Sangmi S. Park, Christian Mueller, Itsaso Garcia-Arcos, Patrick Geraghty.

**Methodology:** Sangmi S. Park, Michelle Mai, Magdalena Ploszaj, Huchong Cai, Itsaso Garcia-Arcos, Patrick Geraghty.

**Resources:** Christian Mueller, Itsaso Garcia-Arcos, Patrick Geraghty.

**Supervision:** Itsaso Garcia-Arcos, Patrick Geraghty.

**Validation:** Itsaso Garcia-Arcos, Patrick Geraghty.

**Visualization:** Sangmi S. Park, Michelle Mai, Magdalena Ploszaj, Huchong Cai, Itsaso Garcia-Arcos, Patrick Geraghty.

**Writing – original draft:** Sangmi S. Park, Itsaso Garcia-Arcos, Patrick Geraghty.

**Writing – review & editing:** Sangmi S. Park, Itsaso Garcia-Arcos, Patrick Geraghty.

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
