## [Decision Letter · Decision Letter 0]

24 Jul 2023

PONE-D-23-20311Type 1 diabetes contributes to combined pulmonary fibrosis and emphysema in male alpha 1 antitrypsin deficient micePLOS ONE

Dear Dr. Geraghty,

Thank you for submitting your manuscript to PLOS ONE. After careful consideration, we feel that it has merit but does not fully meet PLOS ONE’s publication criteria as it currently stands. Therefore, we invite you to submit a revised version of the manuscript that addresses the points raised during the review process.

We look forward to receiving your revised manuscript.

Kind regards,

Doa'a G. F. Al-u'datt

Academic Editor

PLOS ONE

“This work was funded by grants from the Alpha 1 Foundation award numbers 493373 and 614218 (P.G), from the National Heart, Lung, and Blood Institute of the National Institutes of Health under Award Numbers R56HL148774 and R01HL148774 (I. G. A.), and NIH Grants (R01-NS088689, R01-DK098252, and R24-OD018259) and the Alpha-1 Foundation (C.M.). The content is solely the responsibility of the authors and does not represent the official views of the National Institutes of Health or the Alpha-1 Foundation.”

5. We are unable to open your Supporting Information file [Supplemental Fig 1 and Supplemental Fig 2]. Please kindly revise as necessary and re-upload.

Reviewers' comments:

Reviewer's Responses to Questions

**Comments to the Author**

1. Is the manuscript technically sound, and do the data support the conclusions?

Reviewer #1: Partly

Reviewer #2: Yes

Reviewer #3: Yes

Reviewer #4: Yes

Reviewer #5: Partly

2. Has the statistical analysis been performed appropriately and rigorously? 

Reviewer #1: Yes

Reviewer #2: Yes

Reviewer #3: Yes

Reviewer #4: Yes

Reviewer #5: Yes

3. Have the authors made all data underlying the findings in their manuscript fully available?

Reviewer #1: Yes

Reviewer #2: Yes

Reviewer #3: Yes

Reviewer #4: Yes

Reviewer #5: Yes

4. Is the manuscript presented in an intelligible fashion and written in standard English?

Reviewer #1: Yes

Reviewer #2: Yes

Reviewer #3: No

Reviewer #4: Yes

Reviewer #5: Yes

5. Review Comments to the Author

Reviewer #1: The study by Park SS et al is an interesting study investigating comorbidity of Lung disease with experimental diabetes in KO AAT mice model. The concept of the study is very interesting as it address an important issue. However, there are few issues that need to be addressed

a) The scoring of fibrosis needs to be done using the modified Ashrofts score for histology slides and to measure total collagen content of the lung

b) TGFb is one of the most important profibrotic mediator and should be measured

c) possible mechanism should be discussed in details

d) what is the effect on blood sugar level when AAT is administered to the STZ AATKO mice ?

Reviewer #2: The study aimed to investigate the effect of Type 1 diabetes (T1D) on the progression of lung damage in mice with Alpha-1 antitrypsin (AAT) deficiency. The study induced T1D in C57BL/6J background mice using streptozotocin (STZ) and measured pulmonary functions after 3 and 6 months of hyperglycemia. The study also utilized STZ-challenged AAT knockout mice to test the hypothesis. The results suggest that the induction of T1D in AAT deficiency leads to a combined pulmonary fibrosis and emphysema (CPFE) phenotype in male mice. However, the study has some concerns that require addressing by the authors.

1. Introduction:

a. In recent studies, diabetes has been shown to affect the pro-inflammatory environments and proliferative properties of cells. Additionally, it also influences the micro- and macrovascular of the pulmonary vasculature. It is not clear if T1D also affects similarly during the progression of AAT.

2. Methodology:

a. Why were only male mice used in the study?

b. Provide information on the route of STZ administration in mice.

c. How did the authors confirm the AAT KO in mice? Also, it is not clear if the AAT KO was lung-specific or global.

d. Provide the volume of glucose solution injected in mice.

e. Provide details; how much RNA was used for RT-PCR, and how was the genomic contamination confirmed?

f. The format of supplementary figures and information is not accessible.

g. Please explain how the forced expiration volume and forced vital capacity were measured in mice.

3. Results and discussion:

a. Did authors also consider measuring lung resistance and compliance?

b. In Figure 2B, the collagen percentage increased in the lung section of STZ-injected mice. Does it mean the changes in pulmonary function might be associated with airway remodeling mediated due to collagen depositions?

c. Figure 3B, what exactly does the vehicle (40 mg/kg) mean to be here? Also, the dose of STZ mentioned is 40 mg/kg which does not align with the methodology section (50 mg/kg).

Reviewer #3: It is an important study to understand how antitrypsin deficiency contribute pulmonary fibrosis and emphysema in Type 1 Diabetes.

A few things need to be addressed by the authors such as, importantly,

- The introduction and discussion of this study are very brief, authors need to add more background studies and discuss about it in relation to their current study and data, also adding more references in the article.

And also,

- Does the increased levels of collagen in STZ AAT KO mice result in any functional changes? Did the authors confirm it by pulmonary function tests in the later stages? Or how long it does take to appear any lung function abnormalities after the initiation of the collagen formation in these KO mice?

- Is there any specific reason why AAT go down in T1D patients? Or is the diabetic conditions itself causing AAT down regulation in T1D patients?. Such questions need to be discussed elaborately in the manuscript.

- Authors have a Supplementary Figure 2 , but its mention in the manuscript is missing.

- In page 10, line 225-228, Authors mentioned Figure 3B in a Figure 4.?

- In Page 2, line 28-29, the sentence need to be reconstructed.

- In Page 3, line 57-60 , the sentence need to be reconstructed (may be split in to 2 sentences) .

- In Page 9, line 209-210 , the sentence is incomplete , needs reconstruction.

Reviewer #4: This is an interesting study examining the potential role of ATT in STZ diabetes-induced pulmonary consequences. The study concludes that ATT deficiency accelerates pulmonary fibrosis and emphysema compared to WT, when exposed to STZ (starts at 3 months rather than 6 months). The discussion speculates that this effect of ATT may stem from its role regulating EMT transition and WNT signaling and its anti-inflammatory effects. The detrimental effect of knocking out ATT on STZ-induced lung pathology, may also stem from ATT prominent role in the regulation of neutrophil activation and degranulation (review in Janciauskiene et al., 2018), critical in the modulation of inflammation.

Figure 2A,B shows as much increase in collagen staining at 3 months than at 6 months therefore doesn’t reflect the quantification graph in 2A, that shows no difference between Crl and STZ at 3 months.

All the data shown for ATTKO mice reflects the 3 months post hyperglycemia time point. Is there any data at 6 month? Are the fibrosis and PFTs worse at 6 months compared to 3 months? It would be interesting to know, given that the severity of the outcomes is significantly increased in the WT at 6 months.

There are some discrepancies that should be discussed in the mRNA induction of fibrotic markers. Lines 253-255 state that the data shows more pronounced increase in profibrotic markers for the STZ-treated ATTKO compared to the STZ-treated WT, however the following points need to be clarified:

Figure 2A shows no significant change in Acta 2 at 3 months and a significant increase at 6 months (ctl vs STZ) Figure 5B shows only the 3 months data and a significant increase in ATT-KO with or without STZ compared to Ctrl mice. However, is there a significant change between ATTKO ctl vs ATTKO STZ? Is that more pronounced at 6 months? As it is presented, it shows that Acta2 doesn’t change when ATTKO mice are treated with STZ.

Figure 2A also shows significant increases in Ccn2 and Fn1 mRNA in WT STZ diabetic mice at 3 and 6 months but Figure 5B shows a significant increase only in Ccn2 in WT Ctrl vs STZ but not Fn1 as shown in 2A. There are also no significant differences in Ccn2 or Fn1 in the ATTKO mice with or without STZ.

Reviewer #5: This is a very interesting article.

It was known at low doses (usually given as multiple exposure), STZ induce immune and inflammatory response as autoimmune diabetes.

The authors talk about type 1 diabetes, the study was started on adult mice (8-13 weeks).These are older mice, not young ones.Type 1 diabetes is diabetes in young people, usually

The authors should explain this circumstance.

6. PLOS authors have the option to publish the peer review history of their article (what does this mean?). If published, this will include your full peer review and any attached files.

Reviewer #1: No

Reviewer #2: **Yes: **Nilesh Sudhakar Ambhore

Reviewer #3: No

Reviewer #4: No

Reviewer #5: No

---

## [Author Response · Author response to Decision Letter 0]

21 Aug 2023

Response: We have made these changes.

Response: We have updated this section.

“This work was funded by grants from the Alpha 1 Foundation award numbers 493373 and 614218 (P.G), from the National Heart, Lung, and Blood Institute of the National Institutes of Health under Award Numbers R56HL148774 and R01HL148774 (I. G. A.), and NIH Grants (R01-NS088689, R01-DK098252, and R24-OD018259) and the Alpha-1 Foundation (C.M.). The content is solely the responsibility of the authors and does not represent the official views of the National Institutes of Health or the Alpha-1 Foundation.”

Response: We have updated these sections.

Response: We have included a full ethics statement in the resubmitted version.

5. We are unable to open your Supporting Information file [Supplemental Fig 1 and Supplemental Fig 2]. Please kindly revise as necessary and re-upload.

Response: We submitted the supplemental figures as .EPS files and it appears that they were not added to the version given to reviewers. We now include 4 supplemental figures in this resubmission.

Reviewer #1: 

The study by Park SS et al is an interesting study investigating comorbidity of Lung disease with experimental diabetes in KO AAT mice model. The concept of the study is very interesting as it address an important issue. However, there are few issues that need to be addressed

a) The scoring of fibrosis needs to be done using the modified Ashrofts score for histology slides and to measure total collagen content of the lung

Response: We have now performed analysis with the modified Ashcroft score method. Please see Figures 2 and 5B. The data are consistent with the software image analysis.

b) TGFb is one of the most important profibrotic mediator and should be measured

Response: We have performed real-time TGFβ PCR on samples from our 4 groups at the 3-month timeframe and compared the wild type animals at 3- and 6-months post STZ. TGFβ mRNA levels were elevated only in the 3-month wild-type STZ mice. TGFβ was back to baseline in 6-month wild-type STZ mice. Perhaps TGFβ is elevated at an earlier time point in the AAT KO. Please see Figure 6A and where we discuss this on lines 331-351, 378-380, 412-416, and 445-447.

We have also treated human primary lung fibroblasts with recombinant TGFβ in the presence or absence of AAT protein and examined TGFβ-induced ACTA2, CCN2, and FN1 gene expression. AAT reduced TGFβ induction of these fibrosis markers. Please see Figure 6B.

c) possible mechanism should be discussed in details

Response: We have broadened our discussion on possible mechanisms. We have also discussed our new data. Please see lines 390-407, and 412-416.

d) what is the effect on blood sugar level when AAT is administered to the STZ AATKO mice ?

Response: Due to the resubmission deadline, we were unable to perform this experiment. We have outlined this approach in possible future areas for investigation and also outlined previously studies administering AAT to wild type mice and its impact on glucose levels. Please see lines 446-451.

Reviewer #2: 

1. Introduction:

a. In recent studies, diabetes has been shown to affect the pro-inflammatory environments and proliferative properties of cells. Additionally, it also influences the micro- and macrovascular of the pulmonary vasculature. It is not clear if T1D also affects similarly during the progression of AAT.

Response: We added data on several inflammation markers we investigated in 6-month STZ wild-type animals. We observed a significant increase in CXCL5 levels in the plasma of the STZ mice (Supplemental Figure 4). We do observe increased CXCL5 gene expression in the AAT KO mice but this is not enhanced by STZ at the 3-month time point (Figure 6B). Additional time points or analysis of other tissues other than lung may be needed to study the significance of elevated CXCL5. We have included this in our limitation section in the discussion. Please see lines 451-453

We have also outlined this point in the discussion as to whether AAT influences the micro- and macrovascular of the pulmonary vasculature. Please see lines 398-407.

2. Methodology:

a. Why were only male mice used in the study?

Response: Female mice have been found to be resistant to streptozotocin (Bell et al, Endocrinology. 1994; Le May et al, Proceedings of the National Academy of Sciences of the United States of America. 2006; Leiter et al, Proc Natl Acad Sci U S A 1982; Deeds et al, Lab Anim 2011; Rossini et al, Endocrinology 1978). A recent study demonstrated that the resistance of female mice to STZ-induced diabetes can be overcome by increasing the dose of STZ (Saadane et al, PLoS One 2020). We chose to use the established male model to first determine if the lower dose of STZ-induced diabetes would impact the lungs. We have stated this in the discussion on lines 423-426.

b. Provide information on the route of STZ administration in mice.

Response: Mice were intraperitoneally injected with STZ (50 mg/kg) dissolved in citrate buffer for 5 consecutive days. This is now stated on lines 136-137.

c. How did the authors confirm the AAT KO in mice? Also, it is not clear if the AAT KO was lung-specific or global.

Response: We performed ELISA on the plasma of the AAT KO to confirm depletion of AAT. Dr. Mueller’s lab created this line and it is well characterized in his manuscript, Borel et al, Proc Natl Acad Sci U S A. 2018. We have included the screening ELISA data in Supplemental Figure 1. This mouse line is a whole-body knockout.

d. Provide the volume of glucose solution injected in mice.

Response: Glucose solution (250 mg/ml) was injected intraperitoneally at a dose of 2.5 g/kg body weight. Therefore, a 20 g mouse would receive 50 mg of glucose, which would equate to a volume of 250 μl of the 250 mg/ml stock to obtain a 2.5 g/kg. This volume was based on the animal weight. Please see line 136-137.

e. Provide details; how much RNA was used for RT-PCR, and how was the genomic contamination confirmed?

Response: RNA was treated with DNase during the isolation method using the Direct-zol RNA miniprep kit from Zymo Research. RNA (without reverse transcriptase treatment) was tested as a genomic contamination control. 1 ug of total RNA was used for the first strand cDNA template synthesis to generate 20 μl of cDNA. cDNA was dilution by a factor of 4 and 1 μl was used in a 10 μl RT-PCR reaction. Please see lines 173-177.

f. The format of supplementary figures and information is not accessible.

Response: We apologize for the supplemental files not being provided. We did upload them but were not aware that they were not accessible to the reviewers.

g. Please explain how the forced expiration volume and forced vital capacity were measured in mice.

Response: We have added this information. Please see lines 145-150.

3. Results and discussion:

a. Did authors also consider measuring lung resistance and compliance?

Response: Lung resistance and compliance were measure and are shown in Figures 1 and 4. Tissue damping is a measure of tissue resistance and reflects the energy dissipation in the alveoli.

b. In Figure 2B, the collagen percentage increased in the lung section of STZ-injected mice. Does it mean the changes in pulmonary function might be associated with airway remodeling mediated due to collagen depositions?

Response: Yes, we believe that increased collagen likely contributes to the changes in pulmonary function.

c. Figure 3B, what exactly does the vehicle (40 mg/kg) mean to be here? Also, the dose of STZ mentioned is 40 mg/kg which does not align with the methodology section (50 mg/kg).

Response: Thank you for highlighting the error. Yes, 50 mg/kg is the dose of the STZ. We have made this correction in the vehicle group labelling.

Reviewer #3: 

It is an important study to understand how antitrypsin deficiency contribute pulmonary fibrosis and emphysema in Type 1 Diabetes.

A few things need to be addressed by the authors such as, importantly,

- The introduction and discussion of this study are very brief, authors need to add more background studies and discuss about it in relation to their current study and data, also adding more references in the article.

Response: We have added more information to the introduction and discussion as advised. Please see lines 71-80, 390-407, and 412-416.

- Does the increased levels of collagen in STZ AAT KO mice result in any functional changes? Did the authors confirm it by pulmonary function tests in the later stages? Or how long it does take to appear any lung function abnormalities after the initiation of the collagen formation in these KO mice?

Response: We only performed function changes and collagen quantification in AAT KO at the 3-month timepoint post STZ. At 3 months, we do observe collagen changes and function changes in the AAT KO. It would be of interest to test later time point. We have noted this in the limitation section of the discussion. Please see lines 451-453.

- Is there any specific reason why AAT go down in T1D patients? Or is the diabetic conditions itself causing AAT down regulation in T1D patients?. Such questions need to be discussed elaborately in the manuscript.

Response: This is an area for discussion. A previous paper demonstrated that hyperglycaemia can impaired serum AAT concentration and activity (Sandler et al, Diabetes Res Clin Pract. 1988). Typically, AAT is viewed as a marker for inflammation. In T1D, there are higher levels of systemic inflammation. Therefore, the reduced AAT levels observed in our model is consistent with human AAT levels in T1D. The liver is the major site for AAT production. Other cells and tissues produce AAT but the majority comes from the liver. Perhaps hyperglycemia has some long-term effects on liver function that could contribute to lower AAT levels. Equally, it is unclear how hyperglycemia impacts the activity of AAT. These are areas of interest for future studies. Finally, we cannot rule out that STZ itself could impact on liver damage. We have discussed this on lines 434-443.

- Authors have a Supplementary Figure 2 , but its mention in the manuscript is missing.

Response: We apologize for the supplemental files not being provided. We did upload them but were not aware that they were not accessible to the reviewers. 

- In page 10, line 225-228, Authors mentioned Figure 3B in a Figure 4.?

Response: Thank you. We have now corrected this typo.

- In Page 2, line 28-29, the sentence need to be reconstructed.

Response: We have modified this sentence

- In Page 3, line 57-60 , the sentence need to be reconstructed (may be split in to 2 sentences) .

Response: We have modified this sentence

- In Page 9, line 209-210 , the sentence is incomplete , needs reconstruction.

Response: We have modified this sentence

Reviewer #4: 

This is an interesting study examining the potential role of ATT in STZ diabetes-induced pulmonary consequences. The study concludes that ATT deficiency accelerates pulmonary fibrosis and emphysema compared to WT, when exposed to STZ (starts at 3 months rather than 6 months). The discussion speculates that this effect of ATT may stem from its role regulating EMT transition and WNT signaling and its anti-inflammatory effects. The detrimental effect of knocking out ATT on STZ-induced lung pathology, may also stem from ATT prominent role in the regulation of neutrophil activation and degranulation (review in Janciauskiene et al., 2018), critical in the modulation of inflammation.

Response: We agree with the reviewer and have broaden the scope of possible mechanisms that could contribute to this phenotype. The review article from Janciauskiene et al 2018 is a good summary of the literature on AAT regulation on neutrophil responses. We now show elevated levels of CXCL5, a neutrophil chemoattractant, in AAT KO mice and in wild-type STZ injected animals. Please see lines 71-80, 390-407, and 412-416.

Figure 2A,B shows as much increase in collagen staining at 3 months than at 6 months therefore doesn’t reflect the quantification graph in 2A, that shows no difference between Crl and STZ at 3 months.

Response: In Figure 2A the y-axis scale for the percentage of collagen ranges from 0-0.08 %. In figure 2B the y-axis scale ranges from 0-0.25 %. At 3 months the percentage of collagen in the STZ groups are approximately 0.06 % but are 0.17 % at 6 months. Therefore, there is a large difference in collagen from 3 to 6 months. We have adjusted the scale in these figures to be the same range for 3- and 6-month animals. 

Does the reviewer mean that the images look similar in terms of collagen? This is a representative image but the graphs represent many images per animal.

All the data shown for ATTKO mice reflects the 3 months post hyperglycemia time point. Is there any data at 6 month? Are the fibrosis and PFTs worse at 6 months compared to 3 months? It would be interesting to know, given that the severity of the outcomes is significantly increased in the WT at 6 months.

Response: We have not performed AAT KO experiments at the 6-months post STZ. Since we observed significant changes at 3 months, we focused only on this time point. We also expected that the severity of both AAT deficiency and STZ injections might make the animals likely not to survive beyond the 6 months. We have added a sentence in a limitation/future studies section. Please see lines 451-453.

There are some discrepancies that should be discussed in the mRNA induction of fibrotic markers. Lines 253-255 state that the data shows more pronounced increase in profibrotic markers for the STZ-treated ATTKO compared to the STZ-treated WT, however the following points need to be clarified:

Response: We agree with the reviewer and modified this sentence

Figure 2A shows no significant change in Acta 2 at 3 months and a significant increase at 6 months (ctl vs STZ) Figure 5B shows only the 3 months data and a significant increase in ATT-KO with or without STZ compared to Ctrl mice. However, is there a significant change between ATTKO ctl vs ATTKO STZ? Is that more pronounced at 6 months? As it is presented, it shows that Acta2 doesn’t change when ATTKO mice are treated with STZ.

Response: We do not have data on AAT KO mice with and without STZ at the 6-month time point. For Figure 2, we performed Student t-test. While for Figure 5, we performed two-way ANOVA using Tukey’s post hoc tests. There are no significant changes between AAT KO control vs AAT KO STZ for Acta2 gene expression. 

Figure 2A also shows significant increases in Ccn2 and Fn1 mRNA in WT STZ diabetic mice at 3 and 6 months but Figure 5B shows a significant increase only in Ccn2 in WT Ctrl vs STZ but not Fn1 as shown in 2A. There are also no significant differences in Ccn2 or Fn1 in the ATTKO mice with or without STZ.

Response: We agree with the reviewer that not all of these fibrotic markers are enhanced in the AAT KO mice at 3-months post STZ. Additional players likely exist that play a significant role in AAT deficiency associated lung fibrosis in STZ injected mice. Equally, with only sampling at one time point, we may have missed some critical changes that could have occurred earlier. We have discussed this on lines 390-407, 412-416, and 451-453.

Reviewer #5: 

It was known at low doses (usually given as multiple exposure), STZ induce immune and inflammatory response as autoimmune diabetes.

The authors talk about type 1 diabetes, the study was started on adult mice (8-13 weeks).These are older mice, not young ones.Type 1 diabetes is diabetes in young people, usually

The authors should explain this circumstance.

Response: This is an area for debate and many articles discuss possible comparative age ranges between mice and humans. Overall, the maturational rate of mice does not linearly correlate with humans. It is suggested to occur 150 times faster during the first month of life and 45 times faster over the next five months (Flurkey et al 2007). Mature adult mice range in age from 3 - 6 months, which is equivalent for humans ranging from 20 - 30 years. Therefore, 8–12-week-old mice would be deemed to be somewhere close to 15-20 years old in human aging. Also, the standard approach to induced hyperglycemia in C57BL/6J mice is to inject them with STZ between 8-12 weeks old (Jackson labs and others report this). We chose this time range based on established protocols. Please see lines 426-433.

Flurkey, Currer, and Harrison, 2007. 'The mouse in biomedical research.' in James G. Fox (ed.), American College of Laboratory Animal Medicine series (Elsevier, AP: Amsterdam; Boston).

---

## [Decision Letter · Decision Letter 1]

10 Sep 2023

Type 1 diabetes contributes to combined pulmonary fibrosis and emphysema in male alpha 1 antitrypsin deficient mice

PONE-D-23-20311R1

Dear Dr. Geraghty,

We’re pleased to inform you that your manuscript has been judged scientifically suitable for publication and will be formally accepted for publication once it meets all outstanding technical requirements.

Kind regards,

Doa'a G. F. Al-u'datt

Academic Editor

PLOS ONE

Additional Editor Comments (optional):

Reviewers' comments:

Reviewer's Responses to Questions

**Comments to the Author**

1. If the authors have adequately addressed your comments raised in a previous round of review and you feel that this manuscript is now acceptable for publication, you may indicate that here to bypass the “Comments to the Author” section, enter your conflict of interest statement in the “Confidential to Editor” section, and submit your "Accept" recommendation.

Reviewer #1: All comments have been addressed

Reviewer #2: All comments have been addressed

Reviewer #5: All comments have been addressed

2. Is the manuscript technically sound, and do the data support the conclusions?

Reviewer #1: Yes

Reviewer #2: Yes

Reviewer #5: Yes

3. Has the statistical analysis been performed appropriately and rigorously? 

Reviewer #1: Yes

Reviewer #2: Yes

Reviewer #5: Yes

4. Have the authors made all data underlying the findings in their manuscript fully available?

Reviewer #1: Yes

Reviewer #2: Yes

Reviewer #5: Yes

5. Is the manuscript presented in an intelligible fashion and written in standard English?

Reviewer #1: Yes

Reviewer #2: Yes

Reviewer #5: Yes

6. Review Comments to the Author

Reviewer #1: (No Response)

Reviewer #2: The manuscript has been improved from the first version. The authors have addressed my previous comments satisfactorily, I have no further major changes to suggest.

Reviewer #5: (No Response)

7. PLOS authors have the option to publish the peer review history of their article (what does this mean?). If published, this will include your full peer review and any attached files.

Reviewer #1: No

Reviewer #2: No

Reviewer #5: No

---

## [Editor Report · Acceptance letter]

2 Oct 2023

PONE-D-23-20311R1 

Type 1 diabetes contributes to combined pulmonary fibrosis and emphysema in male alpha 1 antitrypsin deficient mice 

Dear Dr. Geraghty:

I'm pleased to inform you that your manuscript has been deemed suitable for publication in PLOS ONE. Congratulations! Your manuscript is now with our production department. 

Kind regards, 

on behalf of

Dr. Doa'a G. F. Al-u'datt 

Academic Editor

PLOS ONE